## [Reviewer comments · BMJ Open]

ARTICLE DETAILS

TITLE (PROVISIONAL)	Mapping the range of policies relevant to care of small and nutritionally at-risk infants under six months and their mothers in Ethiopia: A scoping review protocol
AUTHORS	McGrath, Marie; Woldie, Mirkuzie; Berhane, Melkamu; Abera, Mubarek; Hailu, Endashaw; Rana, Ritu; Lanyero, Betty; Grijalva-Eternod, Carlos; Abdissa, Alemseged; Girma, Tsinuel; Kerac, Marko; Smythe, Tracey

VERSION 1 – REVIEW

REVIEWER	du Plessis, Lisanne Stellenbosch University
REVIEW RETURNED	21-Feb-2023

GENERAL COMMENTS	No comments to submit.
------------------------

REVIEWER	Were, Jason Western University Schulich School of Medicine & Dentistry, Epidemiology and Biostatistics
REVIEW RETURNED	09-Mar-2023

GENERAL COMMENTS	I would like to greatly appreciate the authors for designing a great study focusing on a critical area of concern. Generally, the protocol is well written. Here are my few comments for your consideration: 1. There are a few typos in the document. Further, the authors have used complex sentences in some sections which makes it difficult for the reader to decipher the meaning. For instance, page 4, lines 12-16 "As well as limiting delivery.....". I would recommend a thorough proofreading of the document and reframing of such sentences.2. The authors should also consider providing their full search strategy if they have already completed their searches.3. The strategy of identifying grey literature sounds a bit vague and unreproducible. Are the authors considering conducting forward and backward citation tracking of seminal articles?4. Since the study aims to report 'emerging themes', are the authors intending to perform thematic/content analysis or any other form of analysis? How will they limit bias in such analyses that imputes subjective judgement? The procedure for these analyses should be provided.
--

VERSION 1 – AUTHOR RESPONSE

Reviewer: 1

Prof. Lisanne du Plessis, Stellenbosch University

Comments to the Author:

No comments to submit.

Thank you for taking the time to review.

Reviewer: 2

Mr. Jason Were, Western University Schulich School of Medicine & Dentistry

Comments to the Author:

I would like to greatly appreciate the authors for designing a great study focusing on a critical area of concern.

Thank you for this positive feedback and your suggestions on how to strengthen the study, which I address below.

Generally, the protocol is well written. Here are my few comments for your consideration:

1. There are a few typos in the document. Further, the authors have used complex sentences in some sections which makes it difficult for the reader to decipher the meaning. For instance, page 4, lines 12-16 "As well as limiting delivery.....". I would recommend a thorough proofreading of the document and reframing of such sentences.

Thank you for your detailed review which spotted these. The document has been edited and proofed to correct typos and simplify sentences.

To further aid the reader, I have abbreviated 'small and/or nutritionally at-risk infants under 6 months' to 'at-risk infants under 6 months' in the text, specifying small and/or nutritionally at-risk only when we thought it helpful for the reader to do so (e.g. in the objectives, research questions and when describing the search strategy).

I have also reordered some text, reframed text and simplified language, and added a little more context in the introduction to aid understanding and flow. Three additional references have been added and are highlighted.

I have also corrected a reference to GRADE in Table 1 which was an error – this has been corrected to AGREE II.

2. The authors should also consider providing their full search strategy if they have already completed their searches.

Thank you for this suggestion. The searches have not yet been completed and so a full search strategy is not included (as reflected in response to editor's comments above). The full search strategy will be included in subsequent publication of the completed review.

3. The strategy of identifying grey literature sounds a bit vague and unreproducible. Are the authors considering conducting forward and backward citation tracking of seminal articles?

Thank you for seeking this clarification. Reference lists of all included literature will be additionally hand-searched for any further relevant policy guidance. To aid reproducibility, I have added more detail to the 'Grey literature search' sub-section as follows (see italics for addition):

Grey literature/database searches will be conducted by one researcher and sources and findings cross-checked with a second national researcher. The search process will be mapped and reported.

4. Since the study aims to report 'emerging themes', are the authors intending to perform thematic/content analysis or any other form of analysis? How will they limit bias in such analyses that imputes subjective judgement? The procedure for these analyses should be provided.

Thank you for seeking this clarification. We agree that the term 'emerging themes' is not helpful as it implies a form of analysis beyond what we are planning. As described earlier in the protocol, the MAMI Care Pathway framework and applicable components of the AGREE II (Appraisal of Guidelines for Research and Evaluation) tool will be used to organise and examine details of the policy documents and guide appraisal of their alignment, coherence, opportunities, and gaps. These will guide what we referred to as 'thematic' analysis. I have therefore removed reference to 'emerging themes' in the main text and in the abstract to avoid confusion.

The text has been edited as follows (changes tracked in the amended document):

The MAMI Care Pathway framework and applicable components of the AGREE II (Appraisal of Guidelines for Research and Evaluation) tool will be used to organise and guide appraisal of alignment, coherence, opportunities, and gaps.

Strategic interventions and policy alignment, coherence, opportunities, and gaps will be described in narrative form.

Additional edits to references:

- Existing references have been edited to correct some typos/formatting errors.
- As noted earlier, three new references have been added to support additional text:
 19. Munirul Islam M, Arafat Y, Connell N, et al. Severe malnutrition in infants aged <6 months- Outcomes and risk factors in Bangladesh: A prospective cohort study. *Matern Child Nutr* 2019;15(1):e12642. doi: 10.1111/mcn.12642 [published Online First: 2018/07/05]
 20. Butler S, Connell N, Barthorp H. C-MAMI tool evaluation: Learnings from Bangladesh and Ethiopia. 2018; (58). www.enonline.net/fex/58/cmamitoolevaluation.
 21. Beck K, Kirk C, Bradford J, et al. The Paediatric Development Clinic: A model to improve medical, nutritional and developmental outcomes for high-risk children aged under-five in rural Rwanda. 2018; (58). www.enonline.net/fex/58/thepaediatricdevelopmentclinic.
- A duplicate reference was removed:

Mertens A, Benjamin-Chung J, Colford Jr JM, et al. Child wasting and concurrent stunting in low- and middle-income countries. *medRxiv* 2020 doi: <https://doi.org/10.1101/2020.06.09.20126979>
- Text has been reordered as part of copy-editing and so reference order has changed.

Do not hesitate to contact me for any further clarifications.

VERSION 2 – REVIEW

REVIEWER	Were, Jason Western University Schulich School of Medicine & Dentistry, Epidemiology and Biostatistics
REVIEW RETURNED	28-Apr-2023
GENERAL COMMENTS	Thanks for comprehensively addressing the reviewers concerns. I wish you all the best in executing this study.